# Comparison of Long-Term Strength Development of Steel Fiber Shotcrete with Cast Concrete Based on Accelerator Type

**DOI:** 10.3390/ma13245599

**Published:** 2020-12-08

**Authors:** Kyong Ku Yun, Seunghak Choi, Taeho Ha, Mohammad Shakhawat Hossain, Seungyeon Han

**Affiliations:** 1Department of Civil Engineering, Kangwon National University, 1 Gangwondaegil, Chuncheon 24341, Korea; kkyun@kangwon.ac.kr (K.K.Y.); donghaebi@kangwon.ac.kr (S.C.); gkxogh10@naver.com (T.H.); 2KIIT (Kangwon Institute of Inclusive Technology), Kangwon National University, 1 Gangwondaegil, Chuncheon 24341, Korea

**Keywords:** steel fiber shotcrete, long-term performance, accelerators types, cement mineral accelerator, aluminate accelerator, alkali-free accelerators, tunnel site

## Abstract

This study analyzed the effect of accelerating agents, such as aluminate, cement mineral, and alkali-free accelerators, on the long-term performance of steel-fiber-reinforced shotcrete. The shotcrete performance was studied based on the type and amount of steel fiber added. Performance tests were performed to identify the accelerator providing better long-term performance to the steel-fiber-reinforced shotcrete. Changes in strength and flexural performance over time were investigated. The compressive strength and flexural strength tests on 1-, 3-, 6-, 12-, and 24-month-old test specimens were performed, wherein 37 kg of steel fiber was added to the cement mineral and aluminate mixes, and 40 kg of steel fiber was added to the alkali-free mix. The 1-month compressive strength result of all the test variables satisfied the Korea Expressway Corporation standard. The compressive strength of the cast concrete and shotcrete specimens increased with age, demonstrating a strength reduction, particularly in the 24-month-old shotcrete specimens. Thus, the shotcrete performance may deteriorate in the long-term. In the 24-month-old specimen, substantial flexural strength reduction was observed, particularly in the aluminate and alkali-free specimens. The relative strength of the specimens was compared with that of the cast concrete mold specimens. The results suggest the use of alkali-free accelerators, considering the long-term performance of tunnels and safety of workers. Moreover, increasing the steel fiber performance rather than the amount of low-performance steel fiber must be considered.

## 1. Introduction

Shotcrete is a process in which concrete or mortar is sprayed onto a surface through a hose by compressed air, pneumatically projected, and dynamically compacted under high velocity. Shotcrete is widely used for rock support in tunneling, mining operations, hydropower projects, and slope stabilization owing to its flexibility and rapid strength gain [1]. According to the New Australia Tunneling Model (NATM) construction methods, shotcrete has been extensively used for tunnel construction, and its application has been extended to road, railroad, and subway constructions. Mainly, shotcrete is used in tunnel construction or underground structures to stabilize and protect the structures from land sliding [2].

Shotcrete is used to satisfy basic requirements such as high early strength to provide the possibility of being applied in thick layers without risk of fresh concrete fallout and displacement, particularly in overhead applications. Hence, accelerators are used in shotcrete to develop early strength, reduce rebound, and suppress ground relaxation [3]. In addition, shotcrete accelerators are required to increase the initial strength by promoting concrete setting and hardening, increase the adhesion of concrete and reduce rebound rate and dust, reduce the effect of dust and byproducts on the human body, prevent conveying pipe closure and pulsation. Moreover, accelerators are also required to avoid sudden change due to the variation of the amount added, prevent shotcrete surface exfoliation and sagging phenomenon, provide a low hygroscopic property and good preservability, have less long-term shrinkage and less cracking, and avoid adverse effect such as corrosion of steel [4,5]. Shotcrete accelerators are categorized as aluminate, silicate, alkali-free, and cement mineral accelerators. According to previous studies, an aluminate accelerator is used to improve the cement hydration and its quick setting [6]. In addition, the alkali-free accelerator has a less alkaline content, pH of 0–7, and a long-term strength effect reduction with a low caustic performance [7,8]. Moreover, cement mineral accelerators (C_12_A_7_) were used to estimate shotcrete performance after 28 days for compressive and flexural strength showing enhancement and durability [8,9]. In this study, three types of accelerators are used to evaluate the long-term mechanical performances of shotcrete.

In this research, the shotcrete performance is evaluated by incorporating steel fiber into the mix. The use of fibers to strengthen materials that are weaker in tension than in compression dates back to ancient times where straws were used to reinforce clay bricks [10]. Based on the material they are produced from, fibers are classified as metallic, synthetic, glass, and natural fibers. Synthetic fibers include polyester, polypropylene, and polyethylene. Natural fibers are extracted from wood cellulose, bamboo, and elephant grass. Hybrid fibers are also used to reinforce concrete where more than one type of fiber is added [11]. Metallic fibers, typically steel fibers, are discussed in the ASTM standard considering five types based on the product or the process used for production, such as cold-drawn wire, cut sheet, melt-extracted, mill cut, and modified cold-drawn wire [12,13]. Song [14] investigated the high-strength of steel-fiber-reinforced concrete for different volume fraction, considering 0.5%, 1.0%, 1.5%, and 2.0%. This study mainly focuses on evaluating mechanical properties such as compressive strength, tensile strength, and modulus of rupture. Previous studies showed that a 2.0% volume fraction had better performance [14]. Moreover, Wu et al. [15] examined three types and shapes of steel fiber with different fiber volume percentages that affected the ultra-high-performance concrete mechanical properties. They showed that compressive and flexural strength increased when the fiber content and age increased. Lantsoght [16] studied mechanical properties, such as tensile strength, which was improved. Moreover, crack width and crack spacing were reduced; however, the compressive strength of the concrete was not affected [16]. In addition, Fan et al. [17] focused on the in situ observing the corrosion effect of a new innovative method for steel fiber distribution inside the concrete and long-term durability of the steel-concrete composite.

Even though the use of shotcrete for ground support is well established since its introduction to the construction industry, advances in materials, equipment, and procedures continue to expand its application. Modern and high-performance admixtures and accelerators, steel, and synthetic fibers highly improve shotcrete performance. This study aims to analyze the effect of accelerating agents on the long-term performance of steel-fiber-reinforced shotcrete for tunnel construction by the type of accelerator. The effect of steel fiber’s performance and the added fiber amount on shotcrete performance is also studied. In this regard, the long-term performance of shotcrete is tested according to the type of accelerating agent through the compressive and flexural strength tests. The test considered 1, 3, 6, 12, and 24 months old test specimens using cement mineral, aluminate, and alkali-free accelerators with 37 kg steel fiber incorporated in the cement mineral and aluminate mixes and 40 kg of steel fiber incorporated in the alkali-free mix. This study analyzes the long-term effect of shotcrete and concrete regarding the accelerator and the added steel fiber. The study provides insights into which combination is more appropriate, which is the main purpose and novelty of this study.

## 2. Test Process and Methods

This study focuses on the long-term effect of the tunnel shotcrete on job sites based on three types of accelerators which were evaluated, including steel fibers. The test spotlights on evaluating the mechanical properties through the compressive and flexural strength of every cast in situ specimens. The study also attempted to find a mixture combination providing better performance for proper use

### 2.1. Materials

#### 2.1.1. Cement

Portland cement, used in this study, relied on the ASTM C150; thus, the chemical composition of the cement was 61.2% CaO, 20.8% SiO_2_, and 6.3% Al_2_O_3_. In this study, the exanimated values of the specific area and density were 3300 cm^3^/g and 3.25 g/cm^3^, respectively.

#### 2.1.2. Aggregate

The Korean Expressway Corporation standard for shotcrete aggregate was followed in this study. Thus, a maximum of 10 mm coarse aggregate, river sand, and crushed sand fine aggregate were used. Table 1 summarizes the physical properties of fine and coarse aggregates used in each accelerator mix. Figure 1 showed the mixed aggregate gradation curve of the accelerators.

#### 2.1.3. Accelerators

Accelerators have mainly used for increasing the initial strength. They also play a vital role as contributing material to shotcrete. Moreover, accelerators improve the adhering capabilities of the shotcrete in the construction site. In this study, Korean made cement mineral, aluminate, and alkali-free accelerators were used with the physical properties shown in Table 2 below.

Cement mineral accelerators is a power type accelerator mainly consisting of C_12_A_7_. The characteristics of C_12_A_7_ increase the accelerating and hardening properties of the concrete by producing C_2_AH_8_ [18]. Cement mineral accelerators have the advantage of being easily combined in shotcrete with water because they react with water. Besides, the required amount of cement is lower than other accelerators.

NaAlO_2_ and KAlO_2_ are the main components of the aluminate accelerators, which are mainly used when a large thickness (>15 cm) after excavation and high early strength support are required [8].

Based on EN 934-5, a flash setting admixture can be defined as “alkali-free” when its alkali metal content (sodium and potassium) is lower than 1%, which is expressed as the equivalent of Na_2_O (%Na_2_O + 0.658*%K_2_O), and a pH of 0–7 [19]. Alkali-free accelerators improve the safety of workers as they avoid skin burns, loss of eyesight, and respiratory health problems. Moreover, they contribute to environmental protection by reducing the release of harmful components to groundwater from the shotcrete and its rebound.

#### 2.1.4. Steel Fiber

Hooked type 35 mm length steel fiber was used. Aspect ratio, flexural test, and tensile strength tests were performed on the steel fiber used in this mix. All of the results have satisfied the standard of steel fibers for shotcrete [13]. Table 3 shows the physical properties of the steel fiber used.

### 2.2. Mix Design

The Korean Expressway Corporation (2003) provides guidelines for tunnel shotcrete quality and standard shotcrete composition, which is mainly used to prepare the shotcrete mix design. The amount of each accelerator (5% cement mineral, 5% aluminate, and 7% alkali-free) was set based on the usual mixture trends used by the experts at the construction site. Moreover, for the sustainable slump and air content, a high-range water reduction agent was used in the mixture design in this study. Table 4 showed the proper mix design of using materials for shotcrete performance.

### 2.3. Specimen Preparation

In the experiments, all the concrete and shotcrete specimens were produced on the project site, implying that the specimens are from the actual place of the ongoing tunnel construction. Concrete was provided by the ready-mix plant and concrete mixer truck. Moreover, in the shotcrete test, which was performed on-site, a batch plant was used to mix the ingredients. The mixed concrete was carried to the shotcrete pumping machine by a ready-mixed concrete truck. An air compressor was used to supply air pressure to the nozzle from a separate hose. The shotcrete test by steel fiber performance and mixing amount was performed in the lab using the mixing equipment available in the lab. Concrete reference mixes were prepared for each accelerator mix to compare the results. The symbols given for the specimens are CO for concrete mold and SH for shotcrete core specimen.

In this study, CO specimens were manufactured at the job site along with the steel-reinforced concrete, which was transferred from the batching plant by a ready-mixed concrete truck. These specimens did not contain accelerators. Subsequently, concrete was cast to the cylindrical molds (ø 100 mm × 200 mm) for the compressive strength test as per KSF 2405; additionally, 150 mm × 150 mm × 550 mm size concrete beam specimens were produced to evaluate the flexural performance test as per the Korean Express Corporation. Finally, smoothing both sides of the test specimen was performed for conducting the compressive strength test, as per standard KSF 2405.

SH specimens were manufactured at the job site with the steel-reinforced concrete and accelerators, transferred from the batching plant by a ready-mixed concrete truck. These specimens contained accelerators mixture introduced during shooting. Thus, shooting was performed to the shotcrete test panels 250 mm × 600 mm × 500 mm for core compressive strength and 150 × 150 × 550 mm size concrete beam specimens for the flexural performance. Subsequently, the specimens were properly cured in a water tank at the job site and brought to the laboratory for 1 month. However, for the long-term curing, geotextile sheet and polythene were used for protecting the environmental effect on the specimens, which is shown in Figure 2. After that, core cutting maintained a diameter of the core of 100 mm for the core compressive strength test. This process was based on KSF2784. The core specimen ratio of length to diameter (L/D) was maintained according to ASTM C42-90 and British Standard Institute (BIS) for SH. Here, L/D was set to 2, fixed in all specimens. For the core compressive test, core cutting occurred in the morning and prepared for the test in the afternoon of the testing days. In addition, one concrete test panel was cast to compare the compressive strength of the shotcrete and concrete specimens.

### 2.4. Slump and Air Content Test

KSF 2402 is the Korean standard for the slump test method of concrete that was performed on the fresh concrete before shooting to determine the properties of the mix. The target slump was 100 mm and was maintained properly. According to the KSF 2421, the air concrete test method for concrete by pressure is the method for determining the entrained air amount in the concrete.

The slump test results showed that CM and AF mixes of 140 mm and 125 mm, respectively, have fulfilled the Korean Expressway Corporation standard of 80–150 mm. Moreover, the AL mix slump result was 160 mm, which was slightly higher than the standard. In all mixes, air content was measured below 5%, showing a slight difference between them. This result is acceptable as no specific regulation for air content exists before shooting; it converges to 2–4% after shooting due to the shotcrete property.

### 2.5. Compressive Strength Test

KSF 2405 is the standard test method for the compressive strength of concrete. In this study, this test was performed using 1, 3, 6, 12, and 24 months age ø 100 × 200 mm cylindrical specimens to determine the compressive strength of the mix for the specified ages by considering three specimens per each mix. For the compressive strength test, the properly cured specimens will be prepared for the test by leveling and smoothing both end surfaces of the specimens to let the load be applied uniformly. The compressive strength of the specimen was calculated by dividing the maximum load carried by the specimen by the average cross-sectional area.

### 2.6. Flexural Strength Test

Flexural performance tests were performed by using 150 × 150 × 550 mm beam specimens based on KS F 2566 (Standard test method for flexural performance of fiber reinforced concrete) on the 1, 3, 6, 12, and 24 months age specimens to determine the flexural strength of the mix.

Flexural strength was determined by using a beam specimen under a third-point loading test apparatus. The specimens were prepared with a width of 150 mm when the steel fiber length was above 35 mm, and the width was reduced to 100 mm if the length of the steel fiber used was less than 35 mm. Moreover, the length of the specimen should be higher than three times the width by more than 50 mm. The flexural strength of a specimen can be calculated as follows [20]:(1)$$f_{r}=\frac{Pl}{bh^{2}},$$
where: *f_r_*—Flexural strength (MPa)

*P*—Maximum load obtained (N)

*l*—Span (mm)

*b*—Width of the failed cross-section (mm)

*h*—Height of the failed cross-section (mm)

## 3. Result and Analysis

### 3.1. Effect of Accelerator Types on the Compressive Strength Results

#### 3.1.1. One-Month Compressive Strength Results

In the 1-month-old compressive strength test result, the compressive strength in all of the accelerator mix for the SH specimens was lower than in the CO specimen. Based on the type of accelerator, higher compressive strength was measured in the CM mix CO specimen. However, the SH specimen compressive strength (29.95 MPa) was lower in the CM mix than in the AL (33.49 MPa) and AF (32.85 MPa) mixes, showing almost similar strength. Compared with the strength of the CO specimen, the SH specimen has a relative strength of 58%, 69%, and 68% in the CM, AL, and AF mix, respectively. Figure 3 shows the compressive and relative strength of the mixes by type of accelerator and specimen.

#### 3.1.2. Three-Months Compressive Strength Results

As concrete strength is expected to increase with age, the compressive strength measured in both CO and SH test specimens was higher than in the 1-month-old compressive strength specimens. the CM mix among the CO specimens showed higher compressive strength, with relatively similar strength in the AL and AF mix specimens. In the SH mix, the AF specimen showed higher strength gain (54.4 MPa) than in the 1-month (32.9 MPa) specimens, representing 88% relative strength in comparison with the CO. It was also higher than the AL and AF mixes 45.9 and 39.3 MPa, respectively. The third-month compressive strength values of the concrete and shotcrete specimens of the three accelerator mixes and the comparison graph are shown in Figure 4.

#### 3.1.3. Six-Months Compressive Strength Results

In general, the 6-month-old CO specimens showed higher compressive strength than the 3-month-old specimens. Whereas, there was a slight strength reduction in the SH mixes except for the AL mix that showed approximately a 5% increase in the compressive strength. The AF SH specimen’s compressive strength decreased by 16% in comparison with the 3-month specimens. The relative strength of the AL, CM, and AF SH specimens in comparison with the CO specimens were 72%, 52%, and 71%, respectively, with the CM mix showing a large strength reduction. Figure 5 shows the comparison of CO and SH specimen’s compressive strength by the accelerator type.

#### 3.1.4. Twelve-Month Compressive Strength Results

In the 12-month compressive strength test, all of the CO specimens presented lower strength than the 6-month-old CO specimens. Moreover, the SH specimens showed approximately 6%, 15%, and 39% increase in compressive strength in the AL, AF, and CM mix, respectively. In addition, the difference in the compressive strength for the CO specimens was low, while the SH specimens have higher relative strength than the 6-month specimens. The compressive and relative strength test result by type of accelerator are shown in Figure 6.

#### 3.1.5. Twenty-Four-Months Compressive Strength Results

In the 24-month-old compressive strength test results, no significant difference was observed in the strength among the accelerator mixes in both the CO and SH specimens; however, the CM CO specimen and AL SH specimen showed a slightly higher compressive strength. Compared with the 12-month results, those of the CO specimens showed a slight strength reduction. Nevertheless, there were approximately 9.7 MPa and 7 MPa strength reductions in the CM and AF SH specimens, respectively. The compressive strength test results in the 24-months specimens and the relative strength of the test specimens by mix type are shown in Figure 7.

### 3.2. Effect of Accelerator Types on Flexural Strength Results

#### 3.2.1. One-Month Flexural Strength Results

The flexural performance test was performed on 1-month-old 150 × 150 × 550 mm beam specimens. Concrete reference mixes were prepared for each accelerator mix to compare the results. The maximum flexural strength was calculated by using the maximum load applied on the beam specimen during the test. In the first-month test result, all of the accelerator specimens showed high flexural strength, which satisfied the Korean Expressway tunnel design standard strength of 4.5 MPa. The AL mix showed higher flexural strength of 7.39 MPa in both the CO and SH specimens and the lower strength of 5.81 MPa was measured in the CM mix SH specimen. The AF mixed SH specimen showed higher flexural strength, 6.31 MPa, than the CO specimen flexural strength, 5.92 MPa. Moreover, the AF presented 107% relative strength, while the AL and CM mix specimens had 86% and 84% relative strength, respectively. The flexural strength and relative strength of the specimens are presented in Figure 8.

#### 3.2.2. Three-Month Flexural Strength Results

In the 3-month-old test, all of the specimens for the three accelerator mixes have also satisfied the standard flexural strength of 4.5 MPa. In general, the flexural strength was lower in all mixes in comparison with the 1-month test results. The AL mix showed the highest flexural strength in the 1-month test; however, in this case, it presented the lowest flexural strength for both CO and SH specimens. The CM and AL mix SH specimens presented higher flexural strength than the CO specimen, while the AF mix SH specimen showed similar strength to the CO specimen with 99% relative strength. Figure 9 shows the flexural strength test results along with the relative strength.

#### 3.2.3. Six-Months Flexural Strength Results

In the 6-month flexural strength test, a considerable strength increase in all accelerator mix specimens arose except for the CM mix SH specimen that showed approximately 9% strength reduction in comparison with the 3-month test results. The AL mix CO specimen showed the highest increase in strength, approximately 33%, and approximately 12% strength gain was obtained in the AF and AL mix SH specimens. In addition, the AF mix SH specimen showed higher strength than the CO specimen with a relative strength of 107%, while there was 98% and 91% relative strength in the CM and AL mix specimens. Figure 10 shows the flexural strength test results along with the relative strength.

#### 3.2.4. Twelve-Months Flexural Strength Results

All of the 12-months-old specimens showed higher flexural strength than the standard of 4.5 MPa. Compared with the 6-month test results, those in both the CO and SH specimens of the accelerator mixes showed a high flexural strength except the CM mix SH specimen, which showed a slight strength reduction. The AL mix SH specimen showed approximately 28% strength gain from the 6-month-old specimen, and it showed higher flexural strength than the CO specimen with 107% relative strength. Figure 11 shows the flexural strength test results.

#### 3.2.5. Twenty-Four-Months Flexural Strength Results

All of the 24-months-old CO and SH specimens showed high flexural strength satisfying the standard 4.5 MPa, and the CO specimens’ flexural strength was higher than the SH specimens in all accelerator mixes. Both the CO and SH specimens of the AF mix showed a slightly high flexural strength, and the SH specimens had 81–86% relative strength in comparison with their corresponding CO specimen. Compared with the 12month test results, those of both CO and SH specimens of the AL and AF mixes flexural strength has reduced, while the CM mix showed strength gain. Approximately 24% and 20% in strength reduction occurred in the AL and AF mix SH specimens, respectively. Figure 12 shows the flexural strength of the test specimens by their mix type with the relative strength.

### 3.3. Comprehensive Review of Shotcrete Long-Term Performance Test Results

This section summarizes the 1–24-month-old compressive strength test and flexural performance test results for each accelerator mix to discuss the change in the performance of concrete and shotcrete specimens by type of accelerating agents.

#### 3.3.1. Compressive Strength Test Results

The change in compressive strength from the 1- to 24-month-old specimens showed that, for all accelerators, the CO specimen gains strength until 6 months and a relatively small strength reduction of 3–5% in 24 months. Moreover, the CM and AL mix SH specimens’ compressive strength increased in the 12 months and reduced in the 24 months. The AL mix showed the lowest strength reduction of approximately 7%. In contrast, there was approximately a 19% reduction in the CM mix. The strength change in the AF mix was not steady; however, for the 24-month-old specimens, the compressive strength decreased by 7 MPa in comparison with that for the 12-month-old specimens. Figure 13, Figure 14 and Figure 15 shows the change in compressive strength by age in each accelerator mix. Moreover, Figure 16 showed the relative strength based on the same mixture of CO specimens, which showed that initially, the relative strength of AL and AF were relatively the same. In contrast, the CM presented values always lower than the others. However, when the age of the specimen increased, all accelerators mixtures relative compressive strength increased. The AF mix SH specimens’ relative compressive strength was 88%; thus, higher than others. Moreover, at 12 months, the results of these specimens were higher than AL and CM, which was 84%. At 24 months, the compressive strength reduced more for SH specimens than in the 12-month results, and AL and AF mix SH specimens showed the same results, 73%. Based on these results, and considering the advantages of AF such as durability, environmental safety, and workers’ safety, this accelerator is more useful than AL.

#### 3.3.2. Flexural Strength Test Results

In the flexural performance test, both the CO and SH specimens’ flexural strength increased from 3 to 12 months and decreased after 24 months, except for the CM mix, which showed high flexural strength after 24 months in both the CO and SH specimens. The flexural strength reduction in SH was higher than CO; however, the reduction was not substantial in comparison with the compressive strength change. The changes in both compressive and flexural strengths indicate that the accelerating agents reduce the long-term performance of CO. Figure 17, Figure 18 and Figure 19 shows the change in flexural strength by age for each accelerator mix. Moreover, Figure 20 shows the relative flexural strength (%) of SH that was calculated based on the same mixtures when the CO flexural strength was evaluated. For the 1-month-old specimens, the AF mix SH specimens showed the highest average flexural relative strength, 107%, with respect to the AL and AF specimens. For the 3- and 6-month-old specimens, the CM and AL results increased 108% and 109%, respectively, with respect to the self-CO specimens’ flexural strength, already shown through Figure 18, Figure 19 and Figure 20. Moreover, the age of the specimens increased accordingly, and their self-CO specimens decreased.

### 3.4. ANOVA Test Result

#### ANOVA Test of Compressive Strength

A two-way analysis of variance (ANOVA) was conducted to find statical significance. Here, accelerators indicate CM, AL and AF, also duration indicates 1,3,6,12, and 24 months respectively CO and SH value. After that, we evaluated the significant influence of an ANOVA test on compressive strength and flexural strength test result. Moreover, the results of the analysis shown in Table 5 and Table 6. Then, we evaluated the p-value in all cases and found that in most of the cases the p-value was less than 0.05. Thus, all results of the ANOVA test were shown to be statically significant. However, flexural strength accelerators’ p-values in both the CO and SH cases are not statically significant.

## 4. Conclusions

This study evaluated the SH long-term performance using three types of accelerating agents and mixing ratios of steel fiber. In addition, a performance test based on the type and mixing ratio of steel fiber was conducted. This study is summarized below.
In the SH long-term performance test based on the type of accelerating agent, CM, AL, and AF accelerators were used, and 37 kg steel fiber was added to the CM and AL mix, and 40 kg of steel fiber was used in the AF mix. The performance tests were performed on 1-, 3-, 6-, 12-, and 24-month-old specimens to determine the long-term performance of SH. The following conclusions are drawn based on the results.The compressive and flexural strength test results showed that the CO specimens exhibit a higher strength than that of the SH specimens for all types of accelerators, and both specimens demonstrated a significant strength reduction after 12 months. In the 1-month-old compressive strength test, the Korea Expressway Corporation standard of 21 MPa was satisfied for all test variables. The low compressive and flexural strength for the 24-month-old specimen indicates that accelerating agents reduce the long-term performance; however, the compressive and flexural strength results were not steady in AF. Moreover, the AL mixtures showed steady results in terms of compressive and flexural behavior. They also exhibited the highest compressive strength for the SH, which was 47.5 MPa; however, based on the relative strength results in comparison with those in the CO, both AL and AF were the same at 73%. Furthermore, in terms of flexural strength, the SH of the AF mixtures was highly unsteady. According to the 12-month-old specimen results, the AL SH flexural strength was 7.15 MPa, and that for the 24-month-old specimens was 5.44 MPa. Thus, this study showed that a drastic change occurred in AL flexural strength, which was not seen for the tunnel shotcrete. In contrast, the AF SH showed slightly less flexural strength, 6.82 MPa, than that of the AL SH. However, after 24 months, the flexural strength was slightly high, 5.47 MPa, indicating a greater load-carrying capacity in the long-term. Moreover, the AF accelerator is safer than the AL accelerator for the environment and workers due to the absence of alkali content. Thus, AF is concluded to be more useful than AL and CM for tunnel shotcrete based on the aspects discussed above. Also, the statically significant analysis by the ANOVA test showed that most combinations are statically significant.

In conclusion, it is suggested that it is necessary to promote the use of AF accelerators considering the long-term performance of tunnels and the safety of the workers. Furthermore, the use of high-performance steel fibers should also be considered for a better performance of tunnels. In the future, more tests, such as core cutting, flexural toughness, and chemical composition changes in the concrete by hydration should be performed to find appropriate combinations.

## Figures and Tables

**Figure 1 materials-13-05599-f001:**
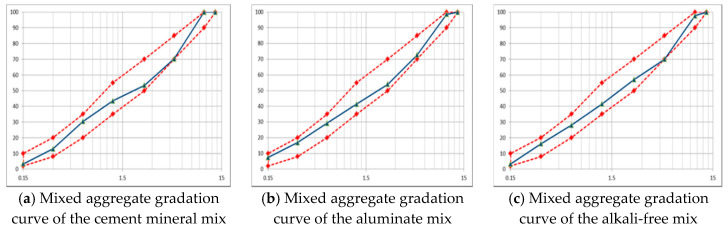
Mixed aggregate gradation curve of the accelerators mix.

**Figure 2 materials-13-05599-f002:**
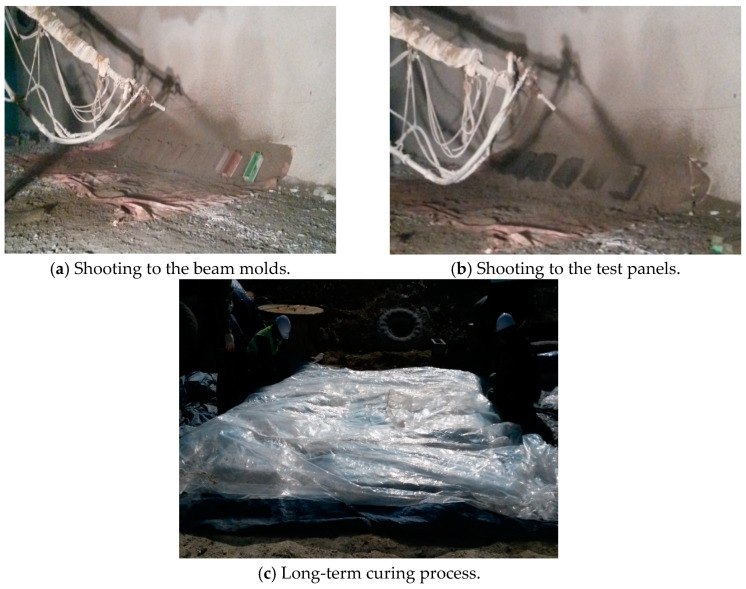
Sample casting and perpetration.

**Figure 3 materials-13-05599-f003:**
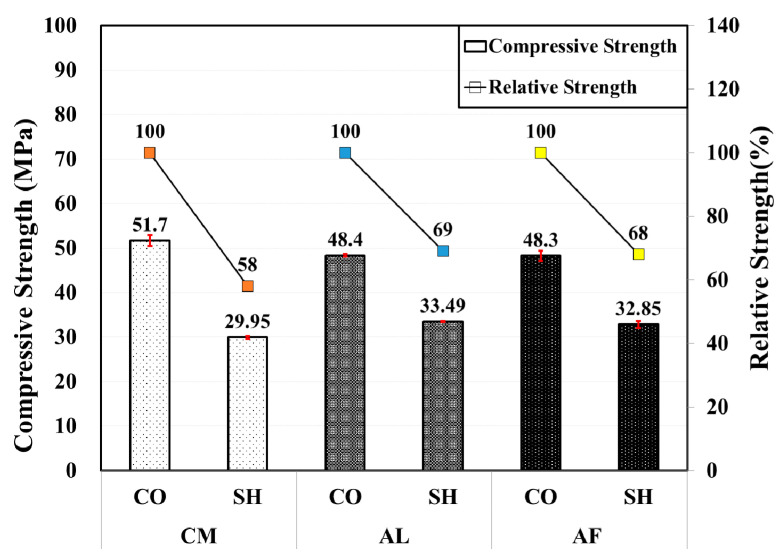
Compressive and relative strength by accelerators for 1-month-old specimens.

**Figure 4 materials-13-05599-f004:**
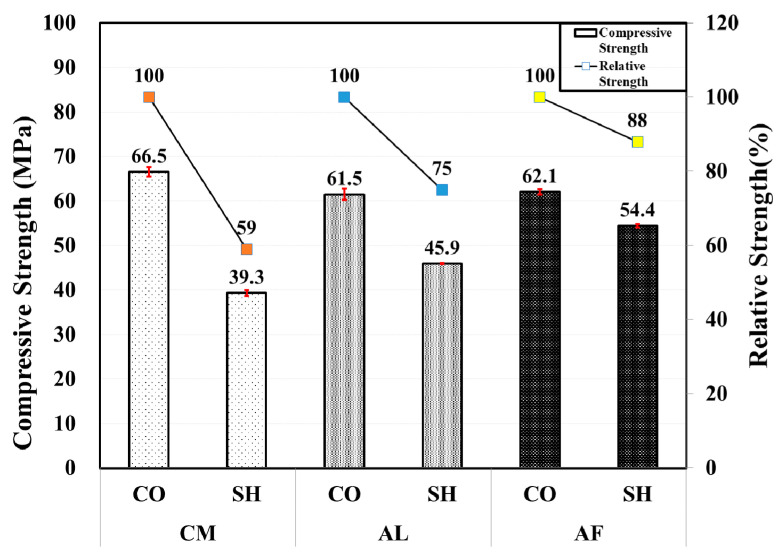
Compressive and relative strength by accelerators for 3-month-old specimens.

**Figure 5 materials-13-05599-f005:**
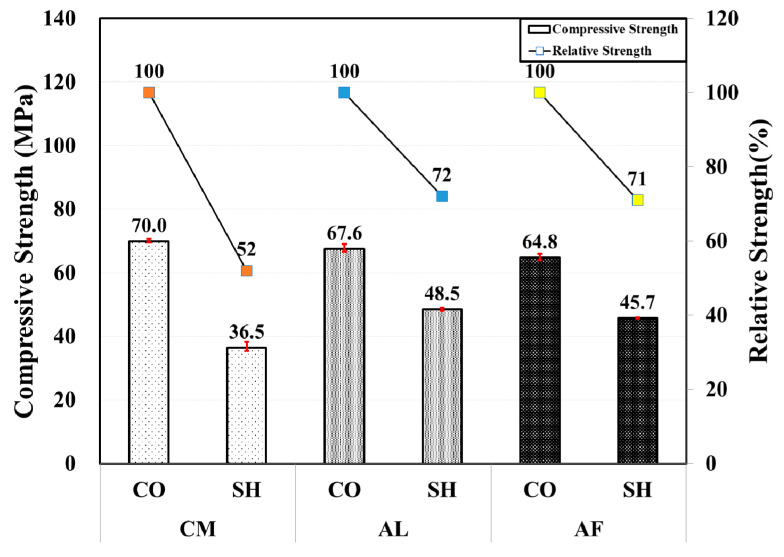
Compressive and relative strength by accelerators for 6-month-old specimens.

**Figure 6 materials-13-05599-f006:**
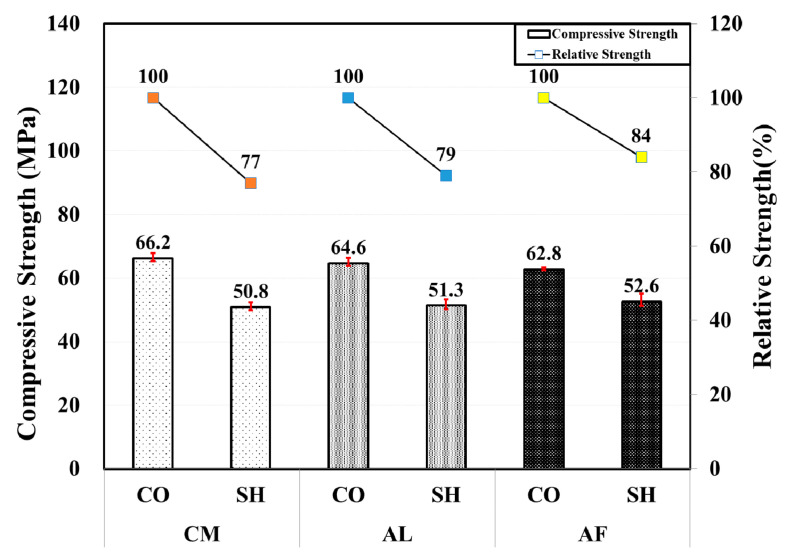
Compressive and relative strength by accelerators for 12-month-old specimens.

**Figure 7 materials-13-05599-f007:**
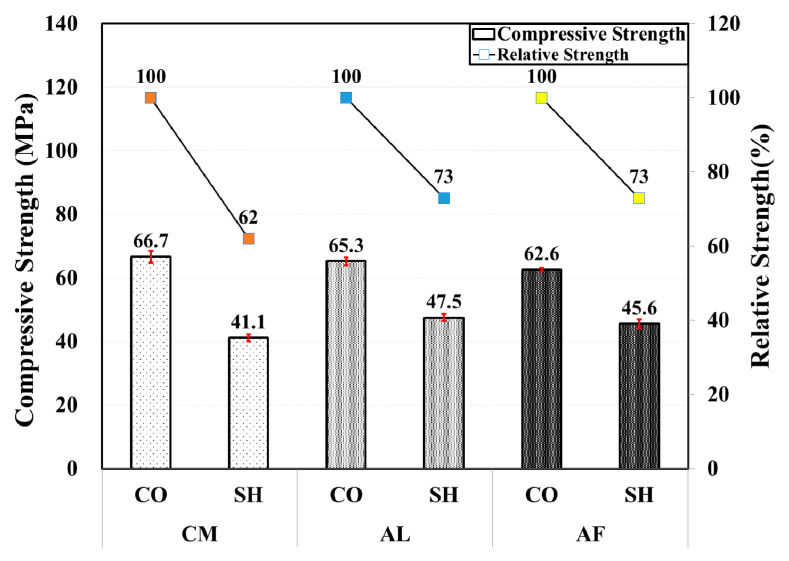
Compressive and relative strength by accelerators for 24-month-old specimens.

**Figure 8 materials-13-05599-f008:**
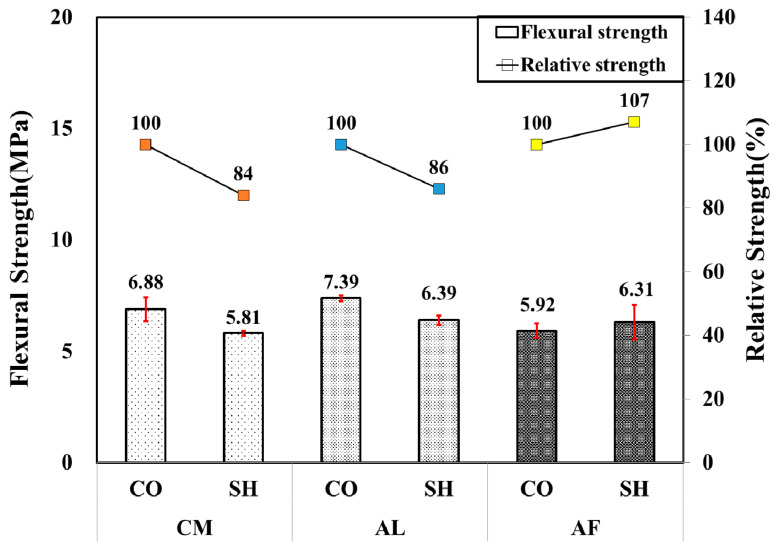
Flexural strength by accelerators for 1-month-old specimens.

**Figure 9 materials-13-05599-f009:**
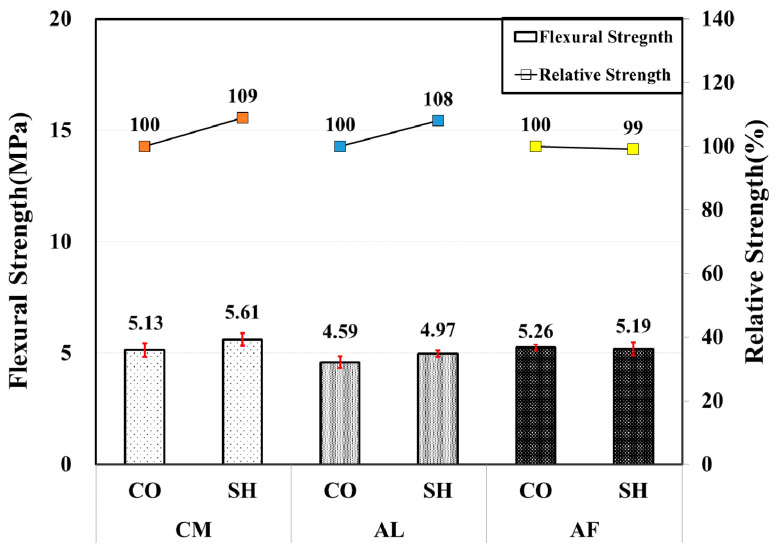
Flexural strength by accelerators for 3-month-old specimens.

**Figure 10 materials-13-05599-f010:**
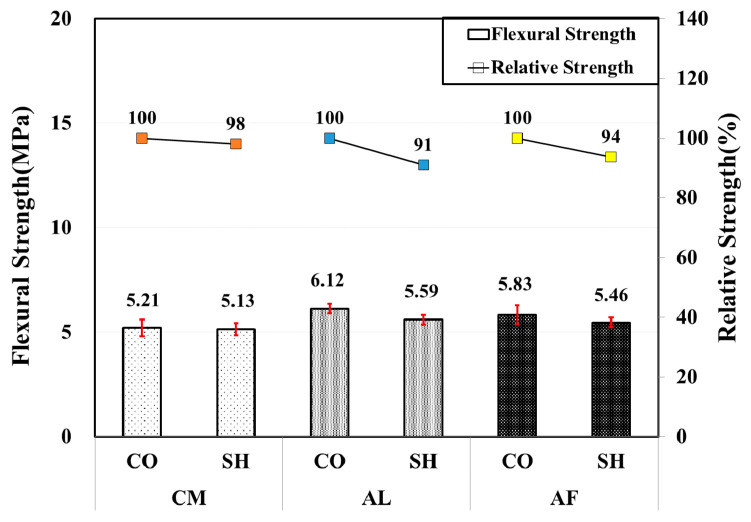
Flexural strength by accelerators for 6-month-old specimens.

**Figure 11 materials-13-05599-f011:**
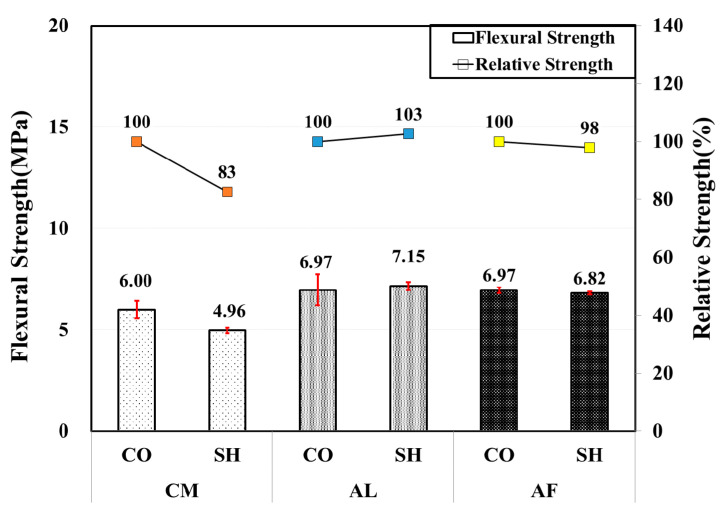
Flexural strength by accelerators for 12-month-old specimens.

**Figure 12 materials-13-05599-f012:**
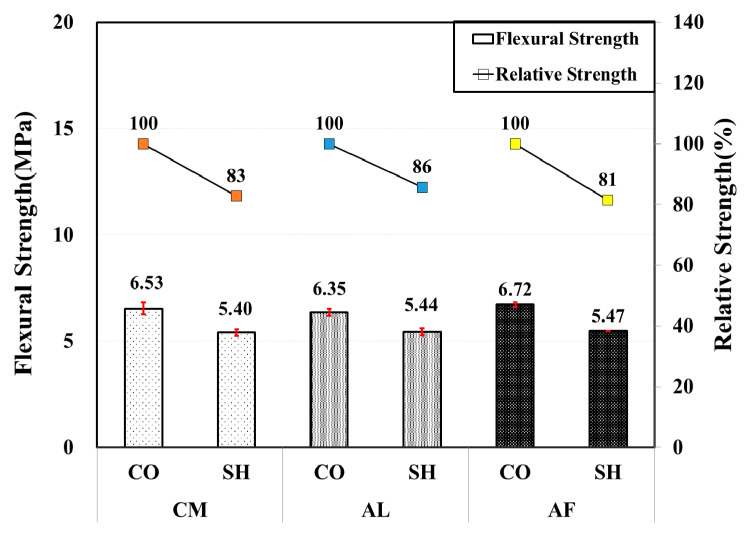
Flexural strength by accelerators for 24-month-old specimens.

**Figure 13 materials-13-05599-f013:**
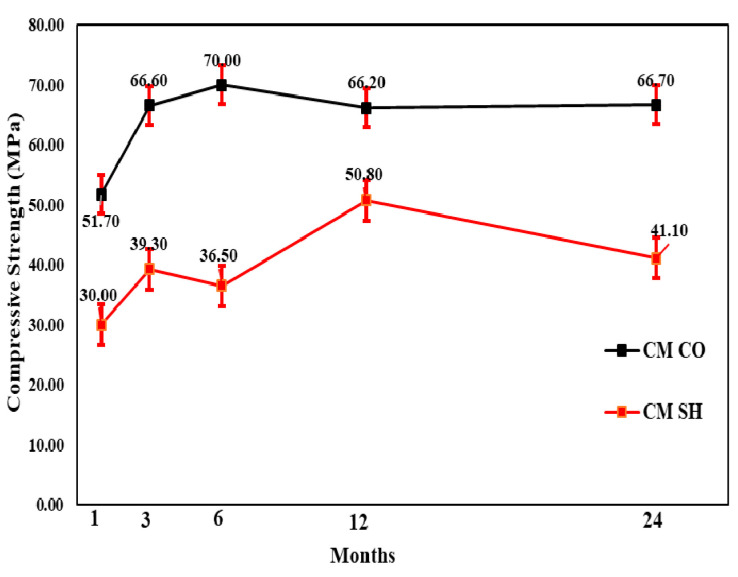
Cement mineral (CM) accelerator mix compressive strength by age.

**Figure 14 materials-13-05599-f014:**
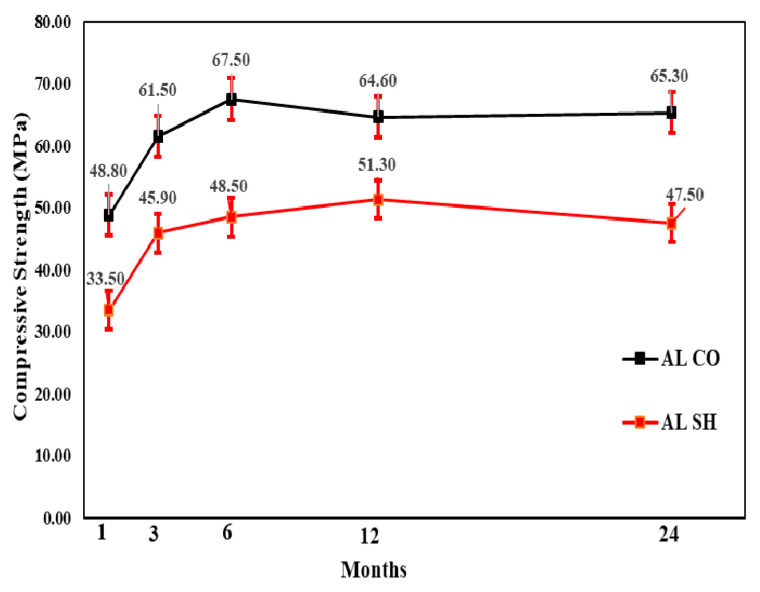
Aluminate (AL) accelerator mix compressive strength by age.

**Figure 15 materials-13-05599-f015:**
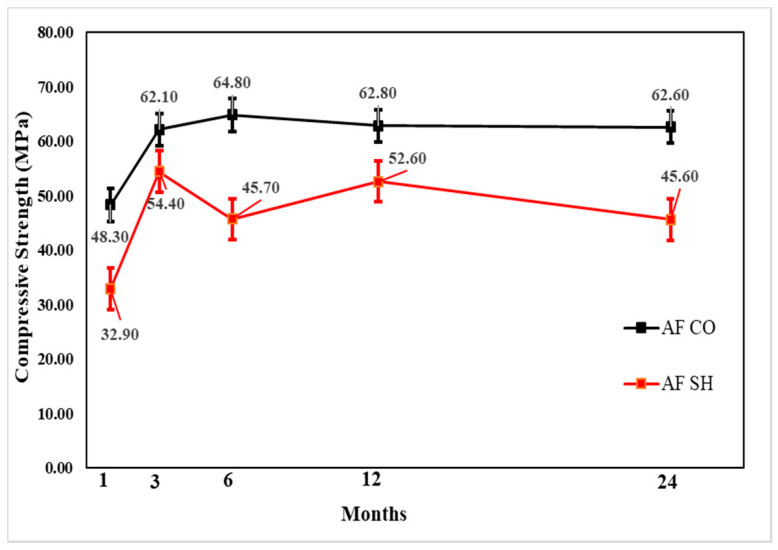
Alkali-free (AF) accelerator mix compressive strength by age.

**Figure 16 materials-13-05599-f016:**
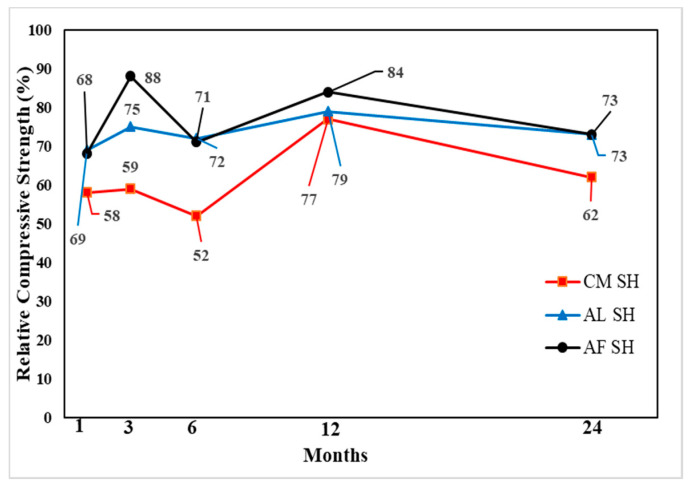
Relative compressive strength (%) by accelerators specimens with shotcrete (SH).

**Figure 17 materials-13-05599-f017:**
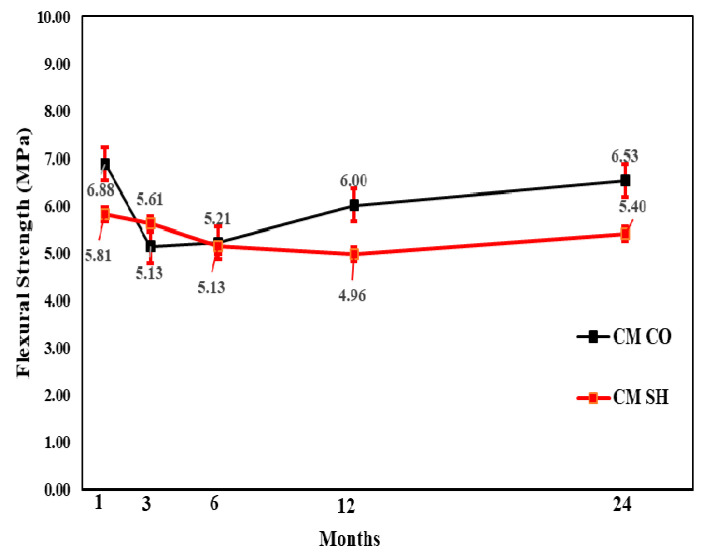
CM accelerator mix flexural strength by age.

**Figure 18 materials-13-05599-f018:**
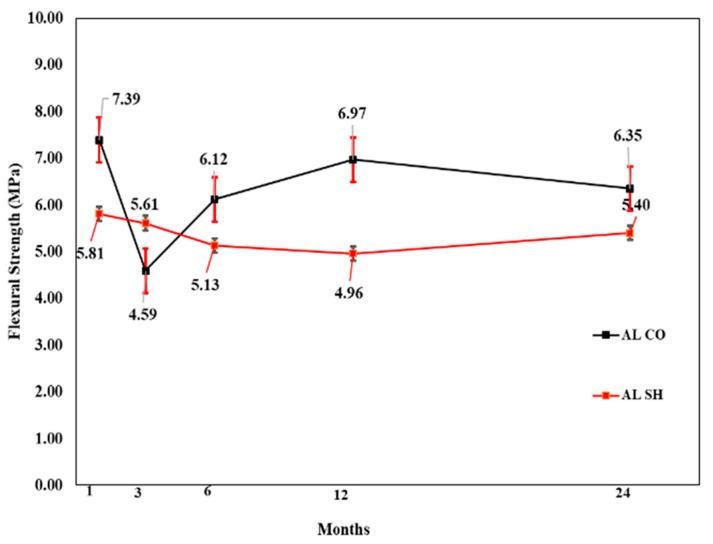
AL accelerator mix flexural strength by age.

**Figure 19 materials-13-05599-f019:**
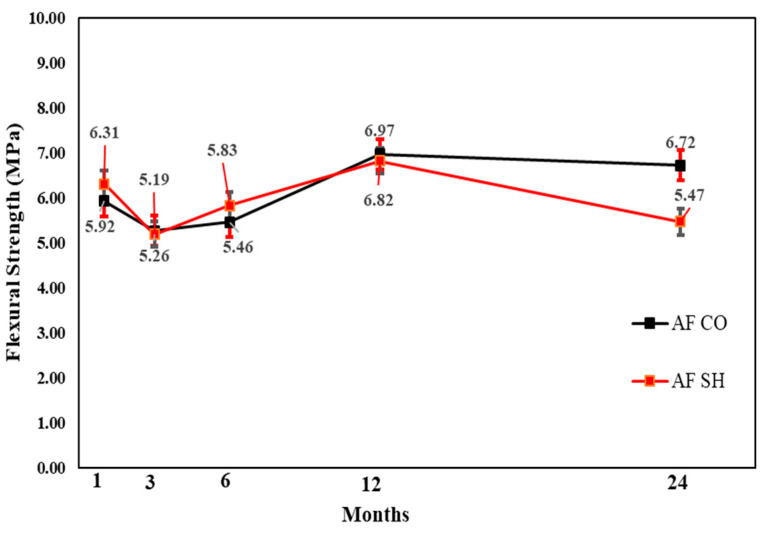
AF accelerator mix flexural strength by age.

**Figure 20 materials-13-05599-f020:**
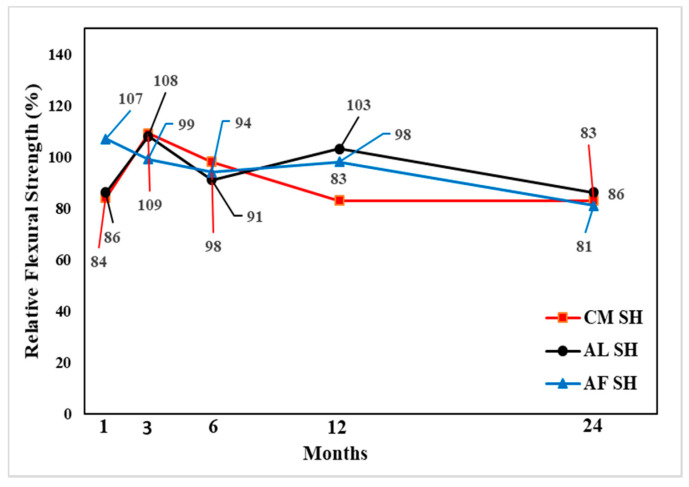
Relative Flexural strength (%) by accelerators specimens with SH.

**Table 1 materials-13-05599-t001:** Physical properties of fine and coarse aggregate.

Test	Aggregate	Density (g/cm^3^)	Fineness Modulus	Division	Standard
Cement mineral mix	Fine	2.61	3.86	Aggregate for shotcrete	Mixed aggregate
	Coarse	2.70			
Aluminate mix	Fine	2.61	3.84		
	Coarse	2.70			
Alkali mix	Fine	2.61	3.78		
	Coarse	2.70			

**Table 2 materials-13-05599-t002:** Physical properties of the accelerators.

Accelerator	Type	Specific Gravity	pH	Solid Content	Initial Set (Hours)	Final Set (Hours)
Cement mineral	powder	2.76	10~12	99.2	2:00	13:00
Aluminate	liquid	1.45	13 ± 2	45.7	3:30	13:20
Alkali-free	liquid	1.36	2.6	42.0	2:45	12:35

**Table 3 materials-13-05599-t003:** Physical properties of the steel fiber.

Test	Material	Aspect Ratio	Tensile Strength
Cement mineral mix	Steel fiber	62.0	1123 N/mm^2^
Aluminate mix	Steel fiber	60.1	1043 N/mm^2^
Alkali-free mix	Steel fiber	60.9	977 N/mm^2^

**Table 4 materials-13-05599-t004:** Shotcrete performance test mix design.

Accelerator	Gmax (mm)	Slump (mm)	W/C	S/a (%)	Unit Content (kg/m^3^)					
					Water	Cement	Sand	Gravel	High-Range Water-Reducing Agent	Steel Fiber (kg/m^3^)
CM *	10	100	0.44	65	210	480	1047	568	4.80 (1.0%)	37
AL *			0.43	64.7	213	492	1074	608	3.936(0.8%)	37
AF *			0.44	62.1	206	465	1011	622	4.65 (1.0%)	40

* CM: Cement mineral. AL: Aluminate, AF: Alkali-free.

**Table 5 materials-13-05599-t005:** Compressive strength analysis of variance (ANOVA) test.

	Mean Squared		F Value		*p*-Value		R^2^	
	CO	SH	CO	SH	CO	SH	CO	SH
Intercept	57598.0167	28654.0906	383.71391	186.563	4.005 × 10^−5^	0.000	0.9896831	0.979
Accelerators	21.6326666	66.1166666	27.035201	5.778	0.0002759	0.028	0.8711141	0.591
Long term	150.106666	153.589	187.59425	13.423	6.139 × 10^−8^	0.001	0.9894512	0.870

**Table 6 materials-13-05599-t006:** Flexural strength ANOVA test.

	Mean Squared		F Value		*p*-Value		R^2^	
	CO	SH	CO	SH	CO	SH	CO	SH
Intercept	562.67367	489.63267	333.497	685.78	0.000	0.000	0.988	0.994
Accelerators	0.14032667	0.41588666	0.507	1.341	0.620	0.315	0.113	0.251
Long term	1.68719333	0.71403333	6.097	2.303	0.015	0.147	0.753	0.535

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
