# Peer review of "Comparison of Long-Term Strength Development of Steel Fiber Shotcrete with Cast Concrete Based on Accelerator Type"

_materials, 2020, doi:10.3390/ma13245599_

Round 1

Reviewer 1 Report

The manuscript presents and discusses the original results concerning the effect of accelerating agents on the long-term performance of steel fiber reinforced shotcrete. From this point of view, the manuscript worth publishing, but only after some moderate revisions.

The main weakness of the manuscript consists of a total absence of a subsection devoted to quality assurance. Indeed, any parameter whose value was experimentally determined must be accompanied by the corresponding uncertainty. 

Moreover, to evidence any statistically significant difference, ANOVA tests should also be used.

I would suggest reducing the number of figures by grouping the existing ones, or, as an alternative, to replace them with a table containing the experimental values, corresponding uncertainties as well as the rest of the ANOVA tests. Only in this way, the authors could prove the significance of their results.

Some small remarks the authors could find on the attached pdf file.

Author Response

  1. Here, I attached the English certificate from editage for proofreading.
  2. In research design, as per your comments I used error bars in all graphs and ANOVA test for statistical significance.
  3. As my analysis, all accelerator types effect incorporates with steel fiber on CO and SH, so that I am thinking individual graphs of long term duration is a proper way to describe the results. Because here, I used relative strength comparison between CO and SH, so accumulate all figures in one position is not fully understandable for the reader. If you have any comments and suggestions about that, I will change it as per comments. Also, I will be glad to consider your valuable comments in this article.
  4. Here I described your all valuable comments which were attached in PDF format.
  • NATM- New Australia Tunneling Model. I used this acronym in the attached file.
  • Page 2. I deleted the repeated word in the study.
  • I deleted the Zemei and used only the family name.
  • Test process and methods: I changed the different patterns of the sentence. So, I am thinking it fulfills your criteria.
  • Also, change all superscripts and subscripts as per your comments.
  • Changed the number of figures 1,2 and 3, and combined one figure with the subtitle a, b, and c.
  • In all figures used error bar as per valuable comment.
  • ANOVA test was added following your comments.

So, I think, my comments are satisfied for fulfilling your review. If you think further revision is needed, I will do that cordially for publishing this paper in your reputed journal.

Reviewer 2 Report

This work deals with the efficiency of some accelerating agents on the long term performance of shotcrete for tunnels and building by studying the mechanical properties. Overall, the MS is clear enough and it falls within the scope of the journal. I recommend some minor revisions as follows:

-Chemical formulas are wrong and they need to be rewritten: 2CaO.Al2O3.8H2O, 12CaO.7Al2O3 and others

-the quality of images should be improved and their position within the manuscript should be optimized. See for instance the captations. Fig. 2 and 3 should be gathered in a single image for better clarity

-the introduction should be updated with the most recent work and literature reports

Author Response

  1. Here, I attached the English certificate from editage for proofreading.
  2. Chemical formulas are wrong and they need to be rewritten: 2CaO.Al2O3.8H2O, 12CaO.7Al2O3, and others. As per your comments, I used C12A7 and C2AH8 instead of this. It is the correct form of chemical formulas.
  3. Changed the number of figures 1,2 and 3, and combined one figure with the subtitle a,b, and c. As per your comments, I changed all figures with error bars and more quality.

4.In the introduction, I mentioned references from 1 to 17. I used 2020, 2019, 2016, and 2015 journal work as a reference.

So, I think, my comments are satisfied for fulfilling your review. If you think further revision is needed, I will do that cordially for publishing this paper in your reputed journal.

Reviewer 3 Report

The manuscript focus on the effect of accelerating agents, such as aluminate, cement mineral and alkali-free accelerators on the long-term performance of steel fiber reinforced shotcrete. In my opinion the manuscript is worth to be published. However, it needs major revision before further processing. My comments are as follows:

- I would like to see the particle size distribution of used aggregates to compare the difference between them,

- I think that the introduction and literature survey should be expanded to highlight the importance of the study properly. It will be especially beneficial to discuss the  tensile and flexural mechanical properties of steel fibre-reinforced concrete (https://journals.sagepub.com/doi/full/10.1177/1464420718782555) or high-strength concrete circular columns made using steel fibers (https://www.mdpi.com/2075-5309/9/10/218) or the effect of the concentration of steel fibres on the properties of industrial floors (https://www.ejournals.eu/Czasopismo-Techniczne/2019/Volume-4/art/14081/),

- Please provide particle size distribution of cement used in this research,

- How the physical properties of fine and coarse aggregate and the accelerators presented in Tables 1 and 2 were determined? Please provide a method of refer to the declaration of the manufacturer,

- To see the scatter of the obtained results please provide error bars on figures from 2 to 11,

- I would like to see some perspectives in conclusion section.

Author Response

Reviewer 3:

  • Here, I attached the English certificate from editing for proofreading
  • First thanks for your positive opinion.
  • Here I attached the particle size distribution for aggregates as per your comments. Here, particle size distribution was different based on accelerators and steel fiber due to different project sites.

Figure 01.Mixed aggregate gradation curve of the cement mineral mix

Figure 01.Mixed aggregate gradation curve of the aluminate mix

Figure 01.Mixed aggregate gradation curve of the alkali-free mix

  • In the paper, I did not attach these figures because these figures are very common and most of the papers are used. So, I think it decreases the quality of the paper as per your reputed journal requirement.
  • The introduction used many references based on accelerators and steel fiber effect on concrete and shotcrete. So, I think it's enough for an introduction and literature survey. Also, I showed the mentioned papers about the citation. These topics I already mentioned here but different citations.
  • Here, we have no particle size distribution of cement.
  • Table 1. physical properties of fine and coarse aggregate were evaluated by on-site aggregate testing as per standard and also in here, I showed the gradation as per your comments.
  • Table 2. Physical properties of the accelerators showed based on the company manufacture process. Here, I shared the link of the company for accelerators.
  • Cement mineral-based accelerators: Union (unioncement.com)
  • Aluminate and alkali-free accelerators: SILKROAD (silkroadcnt.co.kr)
  • I already added error bars for all figures.
  • Also, additionally, I did the ANOVA test.
  • Also, I changed the conclusion of the test result.

So, I think, my comments are satisfied for fulfilling your review. If you think further revision is needed, I will do that cordially for publishing this paper in your reputed journal.

Round 2

Reviewer 1 Report

With the exception of two small suggestions on the attached .pdf I have no more remarks. After the author will correct these misprints, the manuscript could be accepted.

Author Response

Thanks for your opinion. 

As per your comments:

  1. I changed the picture size and leveling.
  2. For ANOVA test result analysis, I did a little bit of sentence pattern changing.

Hopefully, these all fulfill your requirement about the publication in this reputed journal. 

Here also I attached the PDF format of the manuscript.

Reviewer 3 Report

Thank you very much. In my opinion the particle size distribution of aggregates should be presented in the paper to allow others to repeat and valiadate the study.

Author Response

Thanks for your opinion and I put the particle size distribution curve as per your comments. 

Hopefully, it fulfills your requirement.